# FaceCoT: A Comprehensive Benchmark for Face Anti-Spoofing with Chain-of-Thought Reasoning

## Abstract

Face Anti-Spoofing (FAS) typically depends on a single visual modality when defending against presentation attacks such as print attacks, screen replays, and 3D masks, resulting in limited generalization across devices, environments, and attack types. Meanwhile, Multimodal Large Language Models (MLLMs) have recently achieved breakthroughs in image–text understanding and semantic reasoning, suggesting that integrating visual and linguistic co-inference into FAS can substantially improve both robustness and interpretability. However, the lack of a high-quality vision–language multimodal dataset has been a critical bottleneck. To address this, we introduce FaceCoT (Face Chain-of-Thought), the first large-scale Visual Question Answering (VQA) dataset tailored for FAS. FaceCoT covers 14 spoofing attack types and enriches model learning with high-quality CoT VQA annotations. Meanwhile, we develop a caption model refined via Reinforcement Learning (RL) to expand the dataset and enhance annotation quality. Furthermore, we introduce a CoT-Enhanced Progressive Learning (CEPL) strategy to better leverage the CoT data and boost model performance on FAS tasks. Extensive experiments demonstrate that models trained with FaceCoT and CEPL outperform state-of-the-art methods on multiple benchmark datasets.

## 1 Introduction

Face Anti-Spoofing (FAS) plays a vital role in securing face recognition systems, yet it must contend with a wide spectrum of sophisticated presentation attacks such as printed photos, screen-based replay, and 3D masks. The diversity of attack types poses significant challenges to FAS models. However, most existing approaches (Zhou et al., 2022a; Liao et al., 2023; Sun et al., 2023; Le & Woo, 2024) rely solely on a single visual modality, which severely limits their generalization across devices, environments, and attack types, and offers no explicit rationale for their decisions, resulting in poor interpretability.

Meanwhile, Multimodal Large Language Models (MLLMs) (Chen et al., 2024; Bai et al., 2025) have achieved remarkable progress in tasks such as image and text understanding, visual question answering and language reasoning. These models can integrate visual and language information to perform causal reasoning and semantic interpretation, making them particularly suited to FAS tasks. Consequently, a multimodal approach of images and text represents a novel solution path for FAS. Through large-scale pre-training, such models have the potential to overcome the limited generalization of existing FAS methods. Moreover, their strong visual-to-text alignment capabilities combined with powerful language reasoning modules can offer clear decision rationale, thereby significantly enhancing the interpretability of model predictions.

However, this paradigm is hindered by the lack of high-quality image and text multimodal datasets for FAS. Available public FAS datasets (Guo et al., 2022; Boulkenafet et al., 2017; Yu et al., 2020) provide only image or video inputs with binary real or fake labels and omit the structured language information needed for MLLMs training. Using these limited datasets directly for MLLMs training can lead to overfitting and fail to supply an explicit reasoning chain to enhance interpretability. To address this gap, we construct a Chain-of-Thought (CoT) annotated FAS dataset, named **FaceCoT**.

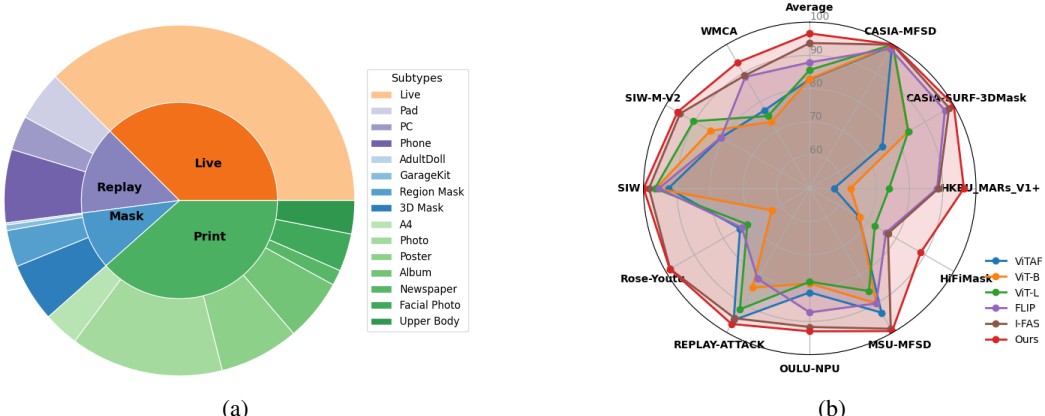

Question: You are a face anti-spoofing expert. Given the following image of a face, answer the question: Is this a spoof?
Answer:
<Caption> In the image, a person is holding a mask with cutouts for the eyes, nose, and mouth, covering their own face. The mask features a printed image of a human face, resembling that of a woman. The background is a white wall, and part of the person's real face, such as some hair, is faintly visible beneath the mask. </Caption>
<Facial description> The mask displays a printed image of a female face, including long wavy hair, a nose bridge, lips, and eye contours. The overall color and appearance are realistic. The cutouts align with the eyes, nose, and mouth, allowing the real face underneath to coordinate with the printed features. </Facial description>
<Facial attributes> Eyes: The printed eye patterns on the mask overlap with the actual eye openings. Nose: The mask has hollowed areas where the nostrils should be. Mouth: There is a hole in the mouth region of the mask. Skin: The mask is made of paper material and covers the person's face. </Facial attributes>
<Reasoning> The real face is hidden behind a mask that uses printed patterns and cutouts to spoof face detection systems. Overall, the combination of the mask's openings and the printed facial features is designed to simulate a real face, including the alignment of features and lifelike textures. Upon close inspection, the face is clearly printed on a flat surface and does not align perfectly with the real face, indicating an attempt to fool the detection system. </Reasoning>
<Spoofing description> Using a printed mask with facial features to obscure the real face. </Spoofing description>
<Conclusion> Yes </Conclusion>

Figure 1: Example of the FaceCoT, illustrating the six CoT components: caption, facial description, facial attributes, reasoning, spoofing description, and conclusion.

(a)          (b)

Figure 2: (a) The data types in FaceCoT. It comprises 3 major spoofing types and 14 subtypes. (b) Comparison results with state-of-the-art methods on 11 benchmark FAS datasets. Our method achieves the highest performance on every evaluation set.

The construction of FaceCoT proceeds in three stages. First, to guide the model to perform human-like reasoning, which involves inspecting from global context down to local details and deriving a final judgment, we design a dedicated CoT annotation format tailored to FAS. As shown in Figure 1, this format comprises six hierarchical levels: caption, facial description, facial attributes, reasoning, spoofing description, and conclusion. Second, we employ GPT-4o (OpenAI, 2023) to automatically generate initial CoT annotations. To guarantee data quality and accuracy, we adopt a human-in-the-loop workflow in which all automatically generated content is rigorously filtered and refined by expert annotators. The resulting high-quality subset is named **FaceCoT-Gold100K**. Third, to expand the dataset's scale, we train a specialized FAS caption model on FaceCoT-Gold100K. During this process, we integrate a rule-verifiable Reinforcement Learning (RL) strategy to improve annotation quality and robustness across domains. The optimized model is capable of producing high-quality CoT annotations even on unseen data, allowing us to generate an additional 982K structured annotations, termed **FaceCoT-Silver982K**.

To the best of our knowledge, FaceCoT is the first Visual Question Answering (VQA) dataset specifically designed for FAS, aggregating **1.08M** training samples and covering 14 distinct attack types. The types of attacks included in the dataset are illustrated in Figure 2(a). Its carefully engineered, structured and hierarchical CoT annotation format not only ensures logical consistency and interpretability but also provides a natural learning pathway for downstream reasoning models. Moreover, we adopt a hybrid annotation workflow that combines GPT-4o–driven automatic generation with expert manual refinement, which further guarantees data quality and reliability. Finally, the integration of an RL–based strategy during the dataset expansion phase enhances cross-domain annotation accuracy and robustness, supplying the multimodal FAS community with unprecedentedly large-scale, high-quality training resources.

The FaceCoT dataset provides a unique resource for training MLLMs. However, if trained in an end-to-end manner, the model is forced to learn CoT reasoning and classification at the same time, which leads to task interference and prevents the reasoning objective from fully converging. As a result, the model cannot fully exploit the fine-grained visual cues embedded in the CoT annotations. To address this, we propose a CoT-Enhanced Progressive Learning (CEPL) strategy, consisting of two stages: (1) Visual Enhancement Pre-training, we perform full-parameter Supervised Fine-Tuning (SFT) of the model using CoT data, thereby focusing the vision encoder on extracting fine-grained, spoof-relevant facial features; (2) Multi-task Joint Training, we inherit the vision encoder from the first stage, reset the connector and language decoder to their original pretrained weights, and fine-tune both the connector and decoder using LoRA modules. We then jointly train on CoT annotations and binary labels with a multi-task loss, ensuring that the model retains deep reasoning capabilities while rapidly adapting to the classification task.

Overall, the contributions of this paper are as follows:

- **Proposing and releasing the FaceCoT dataset.** To address the lack of VQA data in multimodal FAS, we introduce FaceCoT, the first VQA dataset designed specifically for FAS. It covers 14 attack types and comprises 1.08M samples. Each entry contains a structured visual CoT question-answering process, which guides models from image understanding to logical judgment, thereby improving detection accuracy and enhancing generalization.

- **Proposing a CoT-Enhanced Progressive Learning method.** To fully leverage CoT data for FAS detection, we develop a CoT-Enhanced Progressive Learning approach that balances CoT reasoning and binary classification. This strategy significantly improves model performance on both reasoning and classification tasks.

- **Performance.** Extensive experiments on FAS benchmarks demonstrate the value of our dataset and the effectiveness of our method. As shown in Figure 2(b), our approach outperforms state-of-the-art methods, achieving an average AUC improvement of 4.06% and an HTER reduction of 5.00%.

## 2 RELATED WORK

**Face Anti-Spoofing (FAS)** FAS is a key technology for protecting face recognition (Deng et al., 2019) systems from presentation attacks. FAS focuses on detecting these fraudulent attacks to ensure that the faces recognized by the system are genuine live faces. Early FAS methods primarily rely on handcrafted low-level feature extraction techniques, such as LBP (Boulkenafet et al., 2015), SIFT (Patel et al., 2016), and others. With advancements in deep learning, many FAS methods (Liu et al., 2019; Shao et al., 2019) begin using models like CNNs and ViT to train classifiers. FAS methods (Yu et al., 2021; Zhang et al., 2021) gradually introduce multimodal learning, combining features from different types of data (e.g., RGB, IR, depth) to improve model performance. These methods achieve good results in intra-dataset scenarios but struggle with generalization to unseen attack types from out-of-domain data. As a result, Domain Adaptation (DA) (Deb et al., 2023; Cai et al., 2024) and Domain Generalization (DG) (Deb et al., 2023; Guo et al., 2025) approaches are developed. DA minimizes the distribution gap between the source and target domains by utilizing unlabeled target data. However, in practice, collecting unlabeled target data is challenging. DG learns broadly applicable features from multiple source domains, enabling good predictions in the target domain. However, due to the diversity of attack types and data collection methods, it is difficult to find a universal feature space for fake faces, leading to insufficient generalization. Additionally, these methods lack interpretability, making it challenging to understand the decision-making process of the model, which is crucial in high-stakes applications like security systems.

**Chain-of-Thought (CoT) in Multimodal Large Language Models (MLLMs)** In MLLMs, CoT is a method for solving complex problems through a series of intermediate reasoning steps, providing an interpretable reasoning path for the model when answering questions. Recent research (Wei et al., 2022; Wang et al., 2022b) has shown that the CoT prompting method significantly enhances the reasoning abilities of large language models in reasoning tasks. Consequently, researchers have attempted to apply CoT in MLLMs. Some researchers define their own reasoning stages. For example, LLaVA-CoT (Xu et al., 2024) proposes a method combining CoT with MLLMs by designing a 'summary-caption-reasoning-conclusion' reasoning process to enhance the model's reasoning abil-

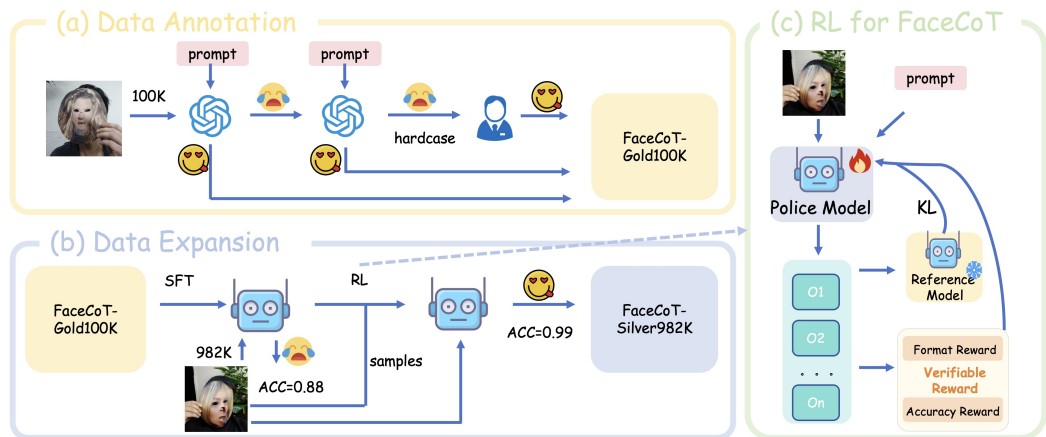

Figure 3: This diagram illustrates the entire process of data annotation and expansion for the Face-CoT dataset. (a) Data Annotation: This step shows the annotation process of FaceCoT-Gold100K. (b) Data Expansion: This phase shows the annotation process of FaceCoT-Silver982K. (c) RL in FaceCoT: This part shows the RL in the training of the FAS caption model.

ity. Other researchers have explored allowing models to autonomously design the reasoning stages, such as PS-CoT (Li et al., 2025), which enables large language models to generate task-solving plans before generating reasoning evidence. CoT-PT (Ge et al., 2023) adopts a hierarchical reasoning approach, moving from abstract concepts to specific ones. BDoG (Zheng et al., 2024) employs a unique approach using three agents who repeatedly debate to implicitly form a reasoning graph that explores and aggregates various thoughts. However, in the field of FAS, such advanced CoT-based reasoning strategies have not yet been fully explored or applied.

## 3 THE FACECOT DATASET

In this section, we introduce the FaceCoT dataset, comprising FaceCoT-Gold100K and FaceCoT-Silver982K, and detail the entire construction process. Figure 3 provides an overview of the pipeline. Subsection 3.1 describes how we collect and curate our raw data to form FaceCoT-Gold100K. Subsection 3.2 provides a detailed explanation of how we design the CoT structure for FAS and use GPT-4o for data annotation. Subsection 3.3 presents the training of our FAS caption model, which is based on FaceCoT-Gold100K under an SFT augmented by RL, and how we then apply it to annotate the complete raw dataset, yielding the larger FaceCoT-Silver982K dataset.

### 3.1 DATA COLLECTION

**Data source** To construct the FaceCoT dataset, our objective is to gather images with large scale, broad attack coverage, and strong demographic diversity to support reliable FAS. We therefore select images that represent both genuine faces and diverse spoofing types, ensuring inclusion of mainstream 2D and 3D presentation attacks under varied conditions. Following these criteria, we collect images from two major FAS datasets: CelebA-Spoof (Zhang et al., 2020), which provides 625K images from 10K subjects with genuine faces and 10 spoofing types, and Wild-FAS (WFAS) (Wang et al., 2023), which contributes 1.38M images (853K spoofed and 529K genuine) from a large number of subjects and 14 attack types captured in unconstrained real-world environments. This combination enables FaceCoT to achieve both high diversity and strong realism.

**Data selection for FaceCoT-Gold100K** In this work, we carefully select samples from the CelebA-Spoof and WFAS datasets to ensure coverage of a broad spectrum of attack types and challenging scenarios. To address this, we divide the selected samples into four main categories: live, replay, print, and mask. First, to ensure data balance across categories, we initially set a target of 25K samples per category. Second, to maximize data diversity, we draw all live-face images exclusively from WFAS, since it is collected under unconstrained conditions and thus contains varied

real-world scenes. For each spoofing category, we exhaustively gather multiple attack styles: for example, print attacks include seven subtypes (e.g., A4, photo, upper body, poster, etc.). In terms of sample selection, we adhered to the principle of data balance, aiming to ensure that each subtype contains an approximately equal number of samples. Screen-replay and mask attacks are selected following the same balanced, subtype-level sampling principle.

## 3.2 DATA ANNOTATION

**CoT structure**    Humans typically judge authenticity of images by following a hierarchical "global-to-local" reasoning path: they first assess the overall scene, then focus on facial details and attributes, and finally draw a conclusion through logical analysis. To emulate this process, we partition our CoT annotations into six modules, enabling the model to perform human-like detection reasoning:

1. **Caption**: First, the model is asked to provide a caption of the entire image. This helps the model understand the global and environmental context of the image and capture more macroscopic spoofing features.

2. **Facial Description**: This part focuses the model's attention on the facial region and requires a description of the facial features. The face is the most easily simulated area in spoofing attacks, so this section helps the model concentrate on potential spoofing regions.

3. **Facial Attributes**: Further, the model is asked to describe various facial attributes, including facial features, textures, and expressions. By describing these attributes, the model's ability to perceive fine-grained details is enhanced.

4. **Reasoning**: Based on the multi-scale information obtained from the first three parts, this section combines both global and local information for a comprehensive analysis to determine whether spoofing behavior exists in the image.

5. **Spoofing Description**: Based on the reasoning process, this section describes the spoofing features and the spoofing method in the image. This not only improves the detection accuracy but also increases the interpretability of the model.

6. **Conclusion**: This section is the final summary of all the reasoning conducted previously.

Based on the above design approach, we further standardize the annotation format of FaceCoT data, adopting the following structure: `<Caption></Caption>`, `<Facial Description></Facial Description>`, `<Facial Attributes></Facial Attributes>`, `<Reasoning></Reasoning>`, `<Spoofing Description></Spoofing Description>`, `<Conclusion></Conclusion>`, thus providing the model with clear and structured input, which helps improve the accuracy and stability of multimodal reasoning (as shown in Figure 1).

**GPT-4o annotation**    After designing the CoT format for the answers, we employ GPT-4o (OpenAI, 2023) to conduct the annotation process. During annotation, for different spoofing types, we provide corresponding hints (e.g., classification standards such as "photographing a poster constitutes spoofing.") to prevent the model from identifying features but failing to properly control the decision boundaries.

**Hard case handing**    After completing the initial annotation, we compare the extracted labels (obtained via regular expressions) with the original ground-truth labels. A total of 98,976 samples are correctly annotated. For those incorrectly annotated, we perform a second round of annotation using GPT-4o. Samples that remain incorrectly labeled after this second attempt are marked as hard cases, resulting in 581 such instances. For these head cases, professional annotators perform cleaning. The annotators review the original labels, locate the corresponding stages, and correct any deficiencies. Finally, they reassemble a logically coherent CoT text. The detailed hard case cleaning process is provided in the Appendix A.2.

## 3.3 DATA EXPANSION

**FAS caption model with Reinforcement Learning**    Although we assembled a diverse FaceCoT-Gold100K, it still falls short of covering every attack type in real-world FAS scenarios. Expanding the dataset with GPT-4o plus human annotation would be costly and hard to scale. Therefore, we

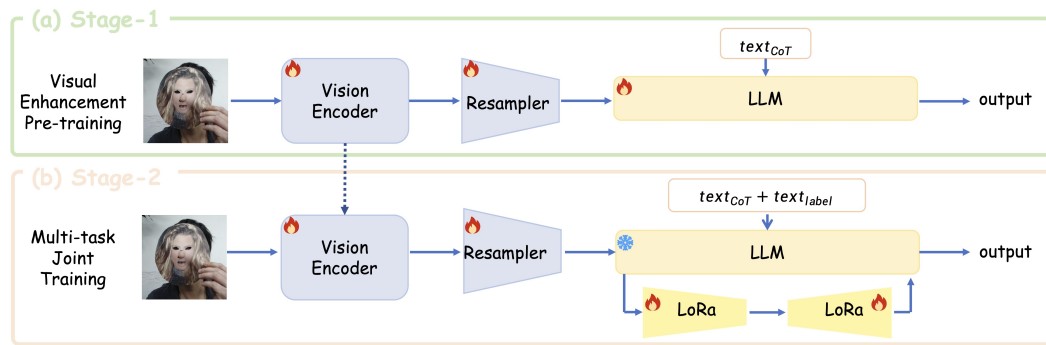

Figure 4: Our proposed CoT-Enhanced Progressive Learning framework consists of two stages: (a) Visual Enhancement Pre-training fine-tunes on CoT annotations to strengthen visual perception and representation; (b) Multi-task Joint Training, which inherits the vision encoder learned in Stage-1 and jointly optimizes both CoT generation and binary classification.

train a caption model to address this issue. We first train a caption model on FaceCoT-Gold100K via SFT, but find that its auto-generated CoT annotations on unseen images suffer from two issues: semantic errors and format errors. To fix this, we adopt a verified reinforcement fine-tuning scheme inspired by VRFT (Liu et al., 2025), designing rewards for both accuracy and format compliance. Specifically, an accuracy reward gives 1 if <Conclusion> matches the image's ground-truth label and 0 otherwise, while a format reward checks whether the output follows the FaceCoT template. This RL step markedly improves annotation correctness and offers a scalable path to enlarging Face-CoT. The details of the RL procedure are provided in the Appendix B.1.

**Caption model annotation** We apply the trained caption model to annotate the training splits of CelebA-Spoof and WFAS. Annotation accuracy is measured by (1) regular-expression checks for template compliance and (2) consistency between the generated <Conclusion> tag and the ground-truth label. With standard SFT, we achieve an accuracy of 88%, whereas reinforcement fine-tuning raises this figure to 99.6%. We ultimately create FaceCoT-Silver982K, an additional 982K high-quality FAS CoT annotations.

## 4 METHODOLOGY

In FAS tasks, the ability to discern fine-grained facial details is critical to model performance. The FaceCoT dataset provides rich fine-grained visual descriptions of facial features, which can effectively guide the learning of a powerful visual encoder. However, adopting a single-stage training strategy typically results in a suboptimal visual encoder. This is because the model is required to learn both reasoning and classification tasks simultaneously, and the faster convergence of the classification objective often leads to insufficient optimization of the reasoning component. Consequently, the model cannot fully leverage the fine-grained visual cues in the CoT annotations, preventing the visual encoder from capturing critical spoofing artifacts and thereby capping overall performance. To fully leverage CoT data, we propose a CoT-Enhanced Progressive Learning (CEPL) strategy. As shown in Figure 4, CEPL consists of two stages: (1) Visual Enhancement Pre-training, we fine-tune exclusively on CoT annotations, which uses language-guided supervision to sharpen the vision encoder's sensitivity to subtle facial features; (2) Multi-task Joint Training, we jointly optimize CoT reasoning and spoof classification to achieve synergy between visual understanding and logical inference.

### 4.1 VISUAL ENHANCEMENT PRE-TRAINING

The goal of this stage is to harness FaceCoT data to enrich the vision encoder's representation of fine-grained facial features, thereby improving cross-modal understanding and laying a solid visual foundation for subsequent Multi-task Joint Training. To this end, we perform full-parameter SFT on CoT data, feeding each image to the model and supervising it with its associated reasoning text.

Table 1: Comparison of evaluation metrics against state-of-the-art methods on 11 benchmark FAS datasets. All baselines are trained on CelebA-Spoof; "Ours-CelebA" is trained on the CelebA-Spoof dataset annotated by the FAS caption model, "Ours-100K" on FaceCoT-Gold100K, and "Ours-All" on the full FaceCoT dataset.

| Methods | CASIA-MFSD | | CASIA-SURF-3DMask | | HKBU-MARs-V1+ | | HiFiMask | |
|---|---|---|---|---|---|---|---|---|
| | HTER(%) | AUC(%) | HTER(%) | AUC(%) | HTER(%) | AUC(%) | HTER(%) | AUC(%) |
| ViTAF (Huang et al., 2022) | 3.11 | 99.48 | 6.18 | 98.40 | 49.29 | 57.28 | 37.30 | 67.10 |
| ViT-B (Radford et al., 2021) | 0.70 | 99.86 | 24.89 | 84.26 | 45.08 | 62.28 | 37.33 | 67.35 |
| ViT-L (Radford et al., 2021) | 0.93 | 99.95 | 23.54 | 84.22 | 33.33 | 73.88 | 32.81 | 72.58 |
| FLIP (Srivatsan et al., 2023) | 4.88 | 98.48 | 8.83 | 96.93 | 17.25 | 88.31 | 28.32 | 76.50 |
| I-FAS (Zhang et al., 2025) | 1.11 | 99.88 | 6.18 | 98.40 | 18.64 | 88.77 | 28.23 | 77.17 |
| Ours-CelebA | **0.00** | **100.00** | 6.21 | 98.73 | **6.96** | **99.41** | 28.68 | 79.74 |
| Ours-100K | **0.00** | **100.00** | 1.33 | 99.79 | 11.74 | 96.37 | 18.63 | 88.52 |
| Ours-All | **0.00** | **100.00** | **0.40** | **99.98** | 7.34 | 98.39 | **15.93** | **91.30** |

| Methods | MSU-MFSD | | OULU-NPU | | Replay-Attack | | Rose-Youtu | |
|---|---|---|---|---|---|---|---|---|
| | HTER(%) | AUC(%) | HTER(%) | AUC(%) | HTER(%) | AUC(%) | HTER(%) | AUC(%) |
| ViTAF (Huang et al., 2022) | 12.86 | 93.14 | 26.73 | 81.28 | 12.38 | 95.73 | 69.34 | 74.22 |
| ViT-B (Radford et al., 2021) | 16.67 | 89.89 | 28.53 | 78.59 | 24.80 | 84.47 | 82.69 | 63.22 |
| ViT-L (Radford et al., 2021) | 20.87 | 85.65 | 29.42 | 78.07 | 16.58 | 92.00 | 80.47 | 71.69 |
| FLIP (Srivatsan et al., 2023) | 19.37 | 89.98 | 20.57 | 87.30 | 25.67 | 81.37 | 80.73 | 73.60 |
| I-FAS (Zhang et al., 2025) | 5.63 | 98.73 | 14.86 | 91.68 | 9.15 | 95.12 | 5.52 | 98.48 |
| Ours-CelebA | 8.33 | 98.29 | 9.47 | 96.01 | 9.87 | 96.84 | **2.45** | **99.66** |
| Ours-100K | **4.58** | **99.56** | 12.70 | 92.99 | **7.37** | **97.07** | 5.51 | 98.57 |
| Ours-All | 5.00 | 99.35 | **5.86** | **97.72** | 12.75 | 95.53 | 4.56 | 99.12 |

| Methods | SIW | | SIW-M-V2 | | WMCA | | Avg. | |
|---|---|---|---|---|---|---|---|---|
| | HTER(%) | AUC(%) | HTER(%) | AUC(%) | HTER(%) | AUC(%) | HTER(%) | AUC(%) |
| ViTAF (Huang et al., 2022) | 14.74 | 92.51 | 26.72 | 80.70 | 29.88 | 77.14 | 23.85 | 82.82 |
| ViT-B (Radford et al., 2021) | 9.13 | 96.24 | 22.60 | 84.59 | 34.72 | 73.10 | 23.48 | 82.98 |
| ViT-L (Radford et al., 2021) | 9.03 | 96.56 | 17.26 | 90.37 | 34.39 | 75.13 | 21.08 | 85.61 |
| FLIP (Srivatsan et al., 2023) | 11.01 | 95.40 | 25.95 | 80.78 | 19.36 | 88.73 | 18.73 | 87.90 |
| I-FAS (Zhang et al., 2025) | 4.02 | 98.34 | 10.89 | 95.02 | 20.07 | 89.17 | 11.30 | 93.71 |
| Ours-CelebA | 0.79 | **99.98** | 9.85 | 96.29 | 11.51 | 95.12 | 8.56 | 96.37 |
| Ours-100K | **0.03** | 99.97 | 9.50 | 95.93 | 12.77 | 93.66 | 7.65 | 96.59 |
| Ours-All | 0.48 | 99.96 | **6.81** | **97.61** | **10.16** | **96.52** | **6.30** | **97.77** |

This process drives precise semantic alignment between visual features and language descriptions, enabling the vision encoder to fully exploit subtle facial cues. As a result, the encoder becomes highly sensitive to spoofing artifacts, providing robust support for the joint optimization that follows.

## 4.2 MULTI-TASK JOINT TRAINING

The objective of this stage is to achieve synergistic enhancement of reasoning and classification capabilities. To this end, we employ a joint training strategy over both CoT reasoning and binary classification tasks. During model initialization, we retain the vision encoder from Visual Enhancement Pre-training to preserve its fine-grained facial representations, while restoring all other modules to their original pretrained weights. We then apply LoRA (Hu et al., 2022) modules to the LLM for targeted fine-tuning. Training proceeds on a combined dataset of CoT-annotated samples and binary labels, guiding the network to balance cross-modal reasoning with accurate spoofing detection. After this stage, the model not only reliably distinguishes real from spoofing faces but also produces coherent CoT explanations for its decisions.

## 5 EXPERIMENTS

### 5.1 EXPERIMENTAL SETTINGS

**Datasets** To validate the robustness and generalization of our data, we use FaceCoT as the source domain and perform cross-domain testing on 11 other datasets as target domains. These datasets include MSU-MFSD (Wen et al., 2015), CASIA-MFSD (Zhang et al., 2012), Idiap Replay Attack (Chingovska et al., 2012), OULU-NPU (Boulkenafet et al., 2017), SIW (Liu et al., 2018b), Rose-Youtu (Li et al., 2018), HKBU-MARs-V1+ (Liu et al., 2018a), WMCA (George et al., 2019), SIW-M-V2 (Guo et al., 2022), CASIA-SURF-3DMask (Yu et al., 2020), and HiFiMask (Liu et al., 2022a). For a fair comparison with previous methods (Zhang et al., 2025), we also replicate their training setup by using only the annotated CelebA-Spoof (Zhang et al., 2020) subset as the source domain while testing on the same 11 benchmarks.

Table 2: Ablation study of the CoT-Enhanced Progressive Learning (CEPL) method. Stage-1: Vision Encoder Pretraining (VEP), Stage-2: Multi-task Joint Training (MJT), Stage-3: Reinforcement Learning (RL). Includes comparisons with single-stage MJT and CEPL variants with/without RL.

Table 3: Ablation study on CoT data, with comparison to a setup using only binary label data of FaceCoT-Gold100K.

| Methods | Stage-1 | Stage-2 | Stage-3 | Results | |
|---|---|---|---|---|---|
| | | | | HTER(%) | AUC(%) |
| Single-stage | - | MJT | - | 8.84 | 95.91 |
| CEPL + RL | VEP | MJT | RL | 7.80 | **96.82** |
| CEPL (Ours) | VEP | MJT | - | **7.65** | 96.59 |

| Data type | Results | |
|---|---|---|
| | HTER(%) | AUC(%) |
| Label | 9.07 | 95.05 |
| Label + CoT | **7.65** | **96.59** |

**Implement details**   We resize all images to $448 \times 448 \times 3$ with RGB channels. We choose MiniCPMV-2.6-8B (Yao et al., 2024) as the backbone VLM for its lightweight multimodal architectureand strong cross-modal fusion capabilities across diverse scenarios; results with alternative backbone VLMs are provided in Appendix C.3. These characteristics make it particularly well-suited to the core demands of FAS tasks, where effective integration of multimodal features is essential. We employ the AdamW optimizer with an initial learning rate of 1e-6 and a weight decay of 0.1. Training is conducted for up to 10 epochs with a global batch size of 256, distributed across 8 NVIDIA A100 GPUs. To ensure robustness, we run all experiments with three different random seeds and report the averaged results. As in previous works (Zhang et al., 2025; Srivatsan et al., 2023; Huang et al., 2022), we use Half Total Error Rate (HTER) and Area Under the Receiver Operating Characteristic Curve (AUC) as evaluation metrics.

## 5.2 RESULTS ANALYSIS

Analyzing the results in Table 1, we can draw three conclusions: (1) Overall performance gain: Training on FaceCoT-All yields an average AUC increase of 4.06% and an HTER reduction of 5.00% over the previous state-of-the-art methods, and it achieves the best score on every evaluation set. (2) Cross-domain generalization: HKBU-MARs-V1+ and HiFiMask include spoofing types absent from the source data(e.g., transparent, plaster, and resin masks). Despite this severe distribution shift, our approach surpasses the previous best AUC by roughly 10% on HKBU-MARs-V1+ and 14% on HiFiMask, demonstrating strong robustness to unseen attack modalities. (3) Single-source comparison: When training is restricted to CelebA-Spoof alone, the model still outperforms the previous state-of-the-art methods: average HTER drops a further 2.74% and AUC rises 2.66%, again securing the top scores on HKBU-MARs-V1+ and Rose-Youtu. This confirms that the CoT annotations and the CEPL training strategy remain effective even under limited-source settings.

## 6 ABLATION STUDY

In this section, we systematically evaluate each major contribution of our framework. All ablation studies are trained on the FaceCoT-Gold100K subset to ensure a consistent comparison baseline.

**Ablation study on the CoT-Enhanced Progressive Learning (CEPL)**   To validate the soundness of our proposed CEPL method, we perform two ablation studies: (i) To validate the effectiveness of the proposed CEPL method, we compare it with the single-stage Multi-task Joint Training method. (ii) Meanwhile, we further extend the CEPL framework by applying RL (Liu et al., 2025) after the second stage, in order to investigate whether RL can bring additional performance gains. As shown in Table 2, we make two key observations: (1) We find that the CEPL method outperforms single-stage training with an increase of 0.68% in AUC and a decrease of 1.19% in HTER. This indicates that stage-wise optimization helps the model better capture the knowledge transfer relationship between different tasks, effectively reducing task interference and improving overall performance. (2) The performance metrics after applying RL are similar to the original method. We hypothesize that in scenarios with limited SFT data, initializing the model with a small amount of SFT and applying RL to unlabeled data can significantly improve performance compared to relying solely on SFT. However, in our study, the model has already undergone extensive SFT with a large amount of high-quality data, resulting in strong baseline performance, so the marginal gains from applying RL afterward are notably reduced.

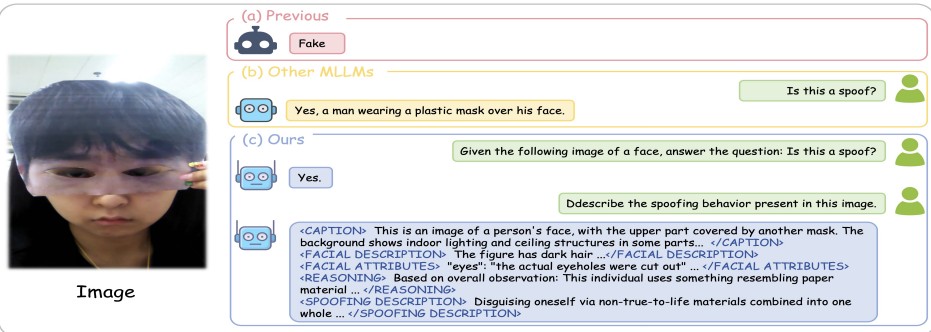

Figure 5: The figure shows the outputs of different FAS methods: (a) Traditional binary classification method; (b) Other MLLMs (Zhang et al., 2025) can answer classifications and provide simple descriptions; (c) Our method can not only answer classification questions, but also provide systematic reasoning analysis.

**Ablation study on FaceCoT**  To validate the effectiveness of the CoT data that we construct during training, we design an ablation experiment at a resolution of $448 \times 448$, comparing single-stage training using only binary-labeled data against training with CoT data under the CEPL framework. The experimental results are shown in Table 3. It can be seen that the model trained with CoT data achieves an HTER of 7.65% and an AUC of 96.59%, both outperforming the model trained without CoT annotations. This result indicates that CoT data provides the model with richer intermediate process information, helping the model better understand complex tasks and significantly enhancing its generalization ability and robustness. Moreover, Figure 5 illustrates that the CoT-trained model not only makes more accurate classification but also exposes its full CoT, especially on attack samples, providing interpretable rationales that further strengthen its cross-domain reliability. This combined evaluation confirms both the performance and interpretability benefits of our constructed CoT dataset.

**Ablation study on resolution**  Capturing fine-grained visual features is critical for FAS detection, and higher input resolution typically provides richer local texture information, thereby enhancing the model's visual representation capacity. To systematically assess the impact of input resolution on model performance, we design a comparative experiment with different resolutions, keeping all other settings consistent. Specifically, we set the input resolutions to $224 \times 224$ and $448 \times 448$, respectively, and compare the performance of the single-stage trained model at both resolutions. The experimental results are shown in Table 4. From the results, it can be seen that when the resolution is $224 \times 224$, the model's performance on all metrics is slightly lower than at the $448 \times 448$ resolution. This suggests that a higher resolution helps the model capture richer detailed features, further enhancing performance.

Table 4:  Ablation study on resolution, comparing input sizes of $224 \times 224$ and $448 \times 448$.

| Resolution | Results | |
|---|---|---|
| | HTER(%) | AUC(%) |
| $224 \times 224$ | 11.28 | 94.05 |
| $448 \times 448$ | **8.84** | **95.91** |

## 7  CONCLUSIONS

We introduce FaceCoT, the first large-scale VQA benchmark dataset for FAS with 1.08 million detailed CoT annotations that cover a wide spectrum of attack types. The dataset begins with 100K high-quality samples, then is expanded to 1.08 million using a reinforcement learning-enhanced caption model. Additionally, we propose a CEPL method that achieves effective synergy between semantic guidance and attack type discrimination. Extensive experiments confirm both the value of the FaceCoT and the effectiveness of our method, delivering an average AUC improvement of 4.06% over current state-of-the-art approaches. We believe that the open-source release of the FaceCoT dataset not only provides a valuable resource for the research community but also lays a solid data foundation and methodological reference for building stronger and more trustworthy FAS systems.

# 8 ETHICS STATEMENT

In designing FaceCoT, attention was paid to fairness, bias mitigation, and data privacy, ensuring that the dataset not only enhances interpretability and generalization in Face Anti-Spoofing (FAS), but also adheres to responsible research principles. To address these potential concerns, this section outlines our efforts in three dimensions: dataset fairness, language model bias mitigation, and data privacy protection.

## 8.1 DATASET BIAS AND FAIRNESS

The original FAS datasets(e.g., CelebA-Spoof (Zhang et al., 2020) and WFAS (Wang et al., 2023)) used in our work were not explicitly designed with fairness auditing or demographic balance in mind. The goal of FaceCoT, however, is to introduce a reasoning-based multimodal framework that improves interpretability and generalization in FAS models. This also provides a structured avenue for detecting and mitigating bias through Chain-of-Thought (CoT) rationales. To align FaceCoT with responsible research practices and address potential fairness concerns, we incorporated the following safeguards during our data collection and annotation process:

- **No New Image Data Introduced:** All FaceCoT annotations are derived exclusively from publicly available FAS datasets. We did not collect or distribute any new images, and the released dataset contains only annotations indexed to existing data.

- **Bias-Aware Annotation Pipeline:** *(1) Prompt Design:* CoT generation prompts were carefully crafted to steer the model toward spoof-specific visual cues (e.g., reflection artifacts and cutting marks), while explicitly excluding references to race, gender, age, or identity. *(2) Human-in-the-Loop Filtering:* Expert annotators reviewed all FaceCoT-Gold100K outputs and were instructed to remove any content with identity-based or stereotypical language. *(3) Random Auditing:* Manual sampling and inspection of 5,000 annotations from FaceCoT-Silver982K revealed no evidence of demographic bias or stereotype leakage. *(4) Model Validation Across Subgroups:* Models trained with FaceCoT demonstrated balanced performance across evaluation datasets, including with respect to skin tone and gender, indicating no observable subgroup bias attributable to the annotations. As shown in the outputs across various evaluation datasets in our Appendix F, the model focused on spoofing features related to FAS rather than attributes such as age, gender, or skin tone.

## 8.2 USE OF GPT-4O AND MITIGATION OF LANGUAGE MODEL BIAS

Large foundation models such as GPT-4o may introduce bias. To mitigate such risks, we employed several safeguards:

- **Constrained Use:** GPT-4o was used solely to generate CoT explanations under strict prompt constraints and was never involved in classification or decision-making tasks.

- **Annotation Safeguards:** *(1)* FaceCoT-Gold100K annotations were reviewed and refined by human experts to eliminate any inappropriate or biased language. *(2)* For FaceCoT-Silver982K, we employed a reward model trained to optimize spoof-specific consistency and rule-based constraints rather than open-ended language fluency, reducing the likelihood of inherited bias.

## 8.3 DATA PRIVACY AND CONSENT

FaceCoT also respects individual privacy and consent, particularly when using publicly sourced visual data. While the datasets employed (e.g., WFAS) are released under academic or Creative Commons licenses, we designed the release to be cautious and transparent:

- **No Image Redistribution:** The FaceCoT release contains only annotations and metadata; no image data is redistributed or exposed.

- **Transparent Documentation:** The final dataset release will include: *(1)* explicit documentation of all source datasets and their associated licenses; *(2)* clear usage guidelines requiring downstream users to adhere to original dataset terms and ethical standards.

## 9 REPRODUCIBILITY STATEMENT

To facilitate reproducibility of our work, we provide in the supplementary materials the code used for GPT-4o annotation. Detailed descriptions of our experimental settings, including dataset preparation, hyperparameter configurations, and evaluation protocols, are provided in Section 5.1. In addition, our evaluation detail is elaborated in Appendix B.3. The proposed CEPL algorithm is clearly described in Section 4, and the reward design and implementation details for reinforcement learning are provided in Appendix B.1. The training and reinforcement learning implementations will be released in a subsequent open-source release. Together, these materials will allow researchers to reproduce the results reported in our paper. Furthermore, we will also make our FaceCoT dataset publicly available to the research community. For the purpose of anonymous review, the dataset has been temporarily hosted at `https://kaggle.com/datasets/40f3d0c3d030fe87c055e3f0658125e817b59026efd314afe741adf2362074e7`.

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

# A  FACECOT DATASET: ADDITIONAL DETAILS

## A.1  GPT-4O ANNOTATION DETAILS

Figure 6 illustrates our Chain-of-Thought (CoT) annotation details using GPT-4o (OpenAI, 2023). To guide GPT-4o to generate accurate, detailed, and correctly formatted responses in a human-like reasoning style, we structure the input into five components—image, system prompt, question, hint, and label—which are concatenated and submitted to the GPT-4o API for annotation.

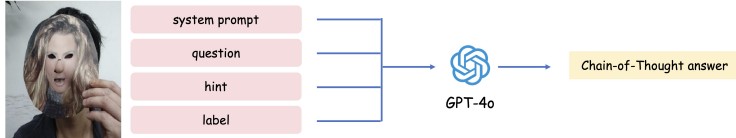

Figure 6: Details of the GPT-4o–based CoT annotation pipeline, with inputs concatenated from five components: image, system prompt, question, hint, and label.

To ensure the consistency and accuracy of our CoT annotations, we apply the following prompt design strategies. First, we explicitly define the model's role as an "examiner" in the system prompt, thereby guiding it to engage in rigorous reasoning from an evaluator's perspective and enhancing both the discriminative power and standardization of its outputs. Second, to enforce structural consistency, we impose an output framework composed of six core modules—caption, facial description, facial attributes, reasoning, spoofing description, and conclusion. Finally, to guarantee accuracy, we require the model's final conclusion to match the standard answer exactly, using this alignment as a key criterion for valid automated annotations. Embedding these requirements at the prompt level effectively codifies the model's reasoning logic and output format, significantly improving result uniformity and batch-processing scalability, while ensuring transparency at every step of the annotation workflow.

---

**System Prompt**

In a face anti-spoofing detection system, a person has submitted an image for inspection. Now please assume the role of the examiner and respond to the question. I need you to strictly follow the format containing six specific parts: caption, facical description, facial attributes, reasoning, spoofing description, and conclusion. It is crucial that you adhere exactly to the outlined structure, and that the final answer in conclusion matches the standard correct answer exactly. Further explanation: In caption, provide a holistic description of the image. In facial description, focus on describing the facial features of the person in the image. In facial attributes, list the facial attributes. In reasoning, perform step-by-step logical reasoning based on the image description, facial description, and facial attributes. In spoofing description, describe the spoofing method and highlight the critical spoofing characteristics in detail. In conclusion, give the final answer \Yes" or \No" that must match the standard answer exactly. Format as follows: <Caption> [Holistic description of the image] </Caption> <Facial Description> [Detailed description of the person's facial features] </Facial Description> <Facial Attributes> [Describe facial attributes: "eyes": ..., "mouth": ...etc.] </Facial Attributes> <Reasoning> [Step-by-step logical reasoning based on the above] </Reasoning> <Spoofing Description> [Details of the spoofing method: ...] </Spoofing Description> <Conclusion> [Answer \Yes" or \No" matching the standard answer exactly] </Conclusion> Please carefully apply this format to analyze the given image and answer the related question.

Furthermore, to prevent the model from detecting spoofing-related features without being able to establish clear decision boundaries (example of such ambiguous case is illustrated in Figure 7), we provide tailored hints for samples of different spoofing types to guide the model toward correct reasoning and conclusions, as presented in Table 5. For spoofing types that the model can readily distinguish, no hints are provided.

Figure 7: Example of a decision-boundary failure: the model correctly detects newspaper features but does not classify the image as spoofing (top). After adding a hint (bottom), the model refines its reasoning and accurately identifies the image as a spoof, demonstrating improved CoT guidance with better decision boundaries.

Table 5: Hints provided for each category; other categories are straightforward and do not require additional hints

| Types | Hints |
|---|---|
| Photo | Photographing a printed photo constitutes spoofing. |
| Newspaper | Photographing a newspaper page constitutes spoofing. |
| Poster | Photographing a poster constitutes spoofing. |
| Album | Photographing an album page constitutes spoofing. |
| A4 | Photographing an A4-printed image constitutes spoofing. |
| Facial print | Photographing a facial photo constitutes spoofing. |
| Upper body | Photographing a printed upper-body image constitutes spoofing. |
| Phone | Photographing a phone screen constitutes spoofing. |
| Pad | Photographing a tablet display constitutes spoofing. |
| PC | Photographing a computer screen constitutes spoofing. |
| Else | - |

## A.2 HARD CASE HANDLING

In the main text, we describe that, following GPT-4o annotation, we apply a regular expression to extract the result within the "<Conclusion>...</Conclusion>" tag and cross-check it against the original label; if the match fails, this annotation is regarded as failed. Samples that could not be correctly labeled after two annotation rounds are designated as "hard cases," resulting in a total of 581 instances. These hard cases are then corrected by human experts. Figure 8 illustrates one such example: experts first verify whether the conclusion is correct, then diagnose why the reasoning and spoofing description are inaccurate; if the reasoning is flawed, they replace it with a correct step-by-step rationale; if the visual feature description is inconsistent, they refine it and iteratively update the subsequent reasoning. Through this expert review and correction process, we ultimately obtain the complete, high-quality FaceCoT-Gold100K dataset. Similarly, samples in the data expansion that fail to be correctly annotated by our caption model are also reviewed and corrected by human experts, leading to the construction of the complete FaceCoT-Silver982K dataset.

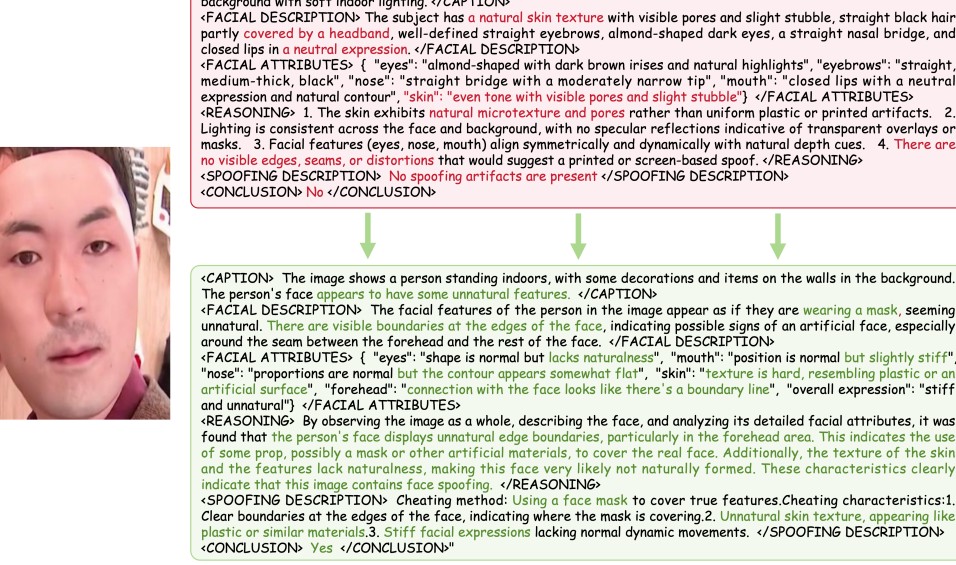

Figure 8: Illustration of hard case handling. The top shows the initial failed annotation, while the bottom presents the revised version by human experts. The subject wears a mask with a clearly visible boundary at the forehead, which is incorporated into the revised annotation.

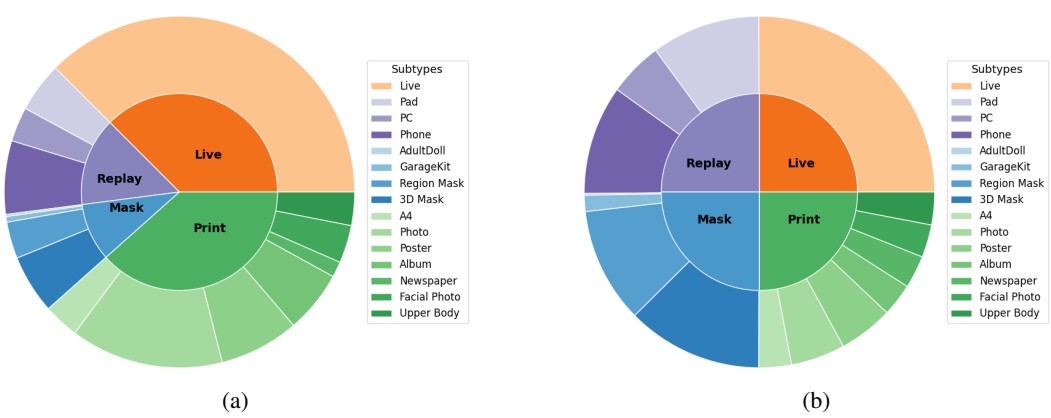

Figure 9: (a) The data types in FaceCoT-Silver982K. (b) The data types in FaceCoT-Gold100K. Both of them comprise 3 major spoofing types and 14 subtypes.

## A.3 STATISTICS

Our FaceCoT dataset comprises two subsets—FaceCoT-Gold100K and FaceCoT-Silver982K—and encompasses living faces alongside 14 distinct spoofing attack types. Here, we present the relative proportions of types across the two datasets in Figure 9 and report the exact sample counts for every category in both subsets in Table 6. Examples of each attack category are illustrated in Figure 10. For the annotation of FaceCoT-Gold100K, we used the GPT-4o API, incurring an average cost of approximately $0.01 per image. To enable large-scale annotation at lower cost, we further trained a caption model via SFT and RL, which required 8 NVIDIA A100 GPUs for about one day, and subsequently employed this model to annotate FaceCoT-Silver982K at a total cost of roughly 288 GPU-hours. Finally, in the human refinement stage, six annotators manually cleaned and verified samples from both subsets over a period of three days.

Table 6: Sample counts per category in the FaceCoT-Gold100K and FaceCoT-Silver982K subsets

| Types | FaceCoT-Gold100K | FaceCoT-Silver982K |
|---|---|---|
| Photo | 5,000 | 138,373 |
| Newspaper | 3,000 | 14,425 |
| Poster | 5,000 | 72,079 |
| Album | 3,000 | 56,490 |
| A4 | 3,000 | 31,776 |
| Facial print | 3,000 | 33,647 |
| Upper body | 3,000 | 30,167 |
| Phone | 10,000 | 66,434 |
| Pad | 10,000 | 48,516 |
| PC | 5,000 | 31,072 |
| 3D mask | 12,768 | 52,637 |
| Region mask | 10,579 | 33,285 |
| Garagekit | 1,488 | 4,505 |
| Adultdull | 165 | 1,454 |
| Living | 25,000 | 367,608 |
| Total | 100,000 | 982,468 |

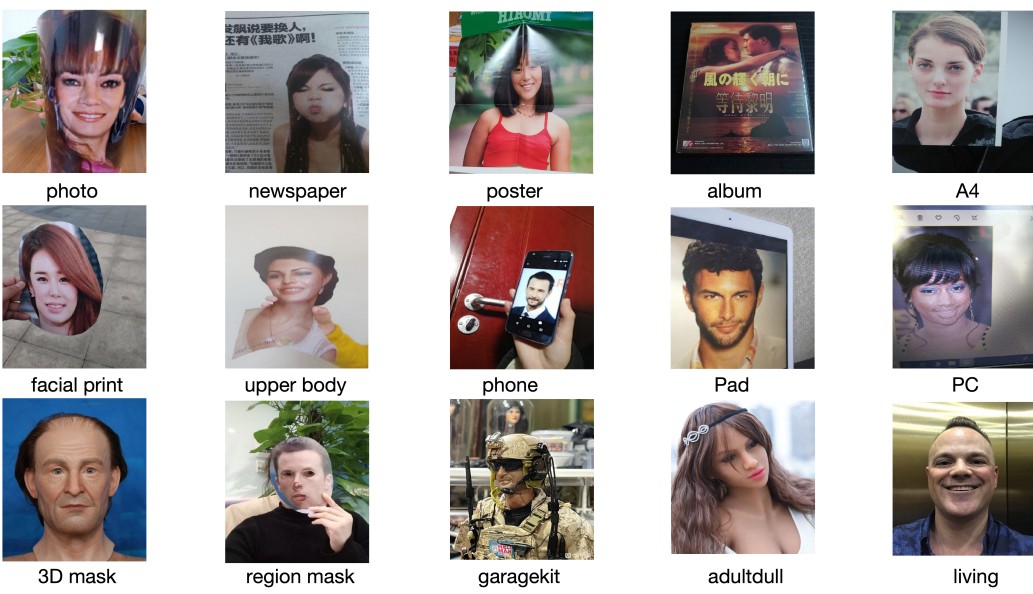

Figure 10: Representative examples of all 14 spoofing attack categories and living faces in the FaceCoT dataset.

# B  METHODOLOGY DETAILS

## B.1  REINFORCEMENT LEARNING IN TRAINING CAPTION MODEL

**Reward functions**  We design a dual-reward scheme targeting both semantic accuracy and output format compliance:

- **Semantic accuracy reward:** Inspired by the ″<Conclusion>...</Conclusion>″ structure in FaceCoT, we apply a regular expression to extract the conclusion from the model's generated output and compare it to the ground-truth label. A match yields a reward of 1; otherwise, 0.

- **Format compliance reward:** We verify whether the model's output follows the prescribed FaceCoT template. If the structural format is correct, the reward is 1; otherwise, 0.

This dual-reward scheme simultaneously enforces correct annotation content and adherence to the FaceCoT formatting guidelines.

**Training strategy**  We initialize the policy model with a version pre-trained via SFT. Given an input image and its associated task prompt, the policy model generates a CoT response. Each response is scored according to the dual-reward functions above, and the resulting reward signal is used to update the policy via RL. To stabilize training, we employ the SFT model as a fixed reference: we compute the KL divergence between the policy's output distribution and that of the reference model, using it as a penalty term to prevent the policy from drifting too far from its initial semantic space. This balance preserves output reliability while enabling effective exploration.

**Training data**  To enhance the caption model's annotation capability and task adaptability on unseen data, we directly use the unlabeled images from the target annotation corpus as input during the RL stage. This construction endows the caption model with strong task-specific adaptation.

**Accuracy evaluation**  We first randomly sample 2,000 instances from the dataset that have not been annotated to construct a test set for evaluation. Then, we use two models to perform automatic annotation on this test set: one trained solely with SFT, and the other further optimized with RL based on the SFT model. From the generated outputs, we extract the result within the ″<Conclusion>...</Conclusion>″ tags and compare them with the original labels of the samples. If the two labels match exactly, the annotation is considered correct; otherwise, it is considered incorrect. The final annotation accuracy is calculated using the following formula:

$$\text{Accuracy} = \frac{\text{Count}(\texttt{conclusion} = \texttt{label})}{\text{Count}(\texttt{conclusion} = \texttt{label}) + \text{Count}(\texttt{conclusion} \neq \texttt{label})} \tag{1}$$

## B.2  REINFORCEMENT LEARNING IN COT-ENHANCED PROGRESSIVE LEARNING (CEPL)

**Motivation**  After the two-stage training with CEPL, the model has demonstrated remarkable anti-spoofing performance on the FAS task. Building upon the success of RL in the caption model, we further investigate its applicability in this component to boost FAS performance while preserving the model's existing capabilities.

**Details**  Specifically, after completing the two-stage training with CEPL, we introduce a third stage of RL. In this stage, we augment the original multi-task loss, which consists of CoT reasoning and classification supervision, with an additional RL objective driven by our dual-reward functions for semantic accuracy and format compliance. The RL procedure follows the same policy-optimization paradigm described previously, with one key difference: no new data is incorporated. Instead, we directly reuse the image–text pairs employed during the second stage. This design tests whether strategic optimization of output structure and semantics, without any additional training examples, can still yield significant performance gains.

### B.3 DETAILS FOR EVALUATION METRICS

Since standard FAS metrics such as AUC and HTER require continuous confidence scores rather than binary predictions, we adapt the output of VLMs to provide probabilistic scores. Specifically:

1. **Deterministic decoding.** To ensure output consistency and avoid randomness from beam search, we set the generation beam number to 1.

2. **Token-level logits extraction.** Instead of directly treating textual outputs (e.g., *yes* vs. *no*) as hard labels, we extract token-level logits from the first generated token. In particular, we identify the token IDs corresponding to 'Yes' and 'No.'

3. **Probability computation.** We compute the softmax probability over the two logits, obtaining the confidence that a sample is *real*:

$$p_{\text{real}} = \frac{\exp(\ell_{\text{No}})}{\exp(\ell_{\text{No}}) + \exp(\ell_{\text{Yes}})}, \tag{2}$$

where $\ell_{\text{Yes}}$ and $\ell_{\text{No}}$ denote the logits of the "Yes" and "No" tokens, respectively.

The resulting probability $p_{\text{real}}$ is then used to calculate AUC and HTER following standard definitions in the FAS literature. This procedure allows us to fairly evaluate LLM-based classifiers under conventional spoofing metrics.

## C EXPERIMENTS

### C.1 CROSS-DOMAIN GENERALIZATION UNDER WIDELY ADOPTED PROTOCOL

In the FAS literature, a widely adopted evaluation protocol is the leave-one-out cross-domain testing on four benchmarks: OULU-NPU (O) (Boulkenafet et al., 2017), CASIA-MFSD (C) (Zhang et al., 2012), Idiap Replay-Attack (I) (Chingovska et al., 2012), and MSU-MFSD (M) (Wen et al., 2015). However, the performance under this protocol has already saturated (with AUC exceeding 99%), making it less discriminative for assessing fine-grained improvements. Therefore, in the main text we focus on a more challenging and generalization-oriented one-to-eleven protocol, which better highlights the advantages of our method. Nevertheless, to further demonstrate the robustness of our approach, we also report results under the widely used leave-one-out protocol. Specifically, we first apply our FAS caption model to generate CoT annotations for the training splits of the O, C, M, and I datasets. Based on these annotated datasets, we then perform four cross-dataset evaluations following the standard protocols. For example, the protocol OCI→M denotes that the model is trained on OULU-NPU, CASIA-MFSD, and Idiap Replay-Attack, and tested on MSU-MFSD. Similarly, OMI→C, OCM→I, and ICM→O are defined in the same manner. As shown in Table 7, our method outperforms previous state-of-the-art methods, achieving the best average HTER and AUC. These results confirm the effectiveness of our approach in improving generalization in FAS.

Table 7: Cross-dataset evaluation results under widely used cross-domain protocol.

| Method | O&C&I→M | | O&M&I→C | | O&C&M→I | | I&C&M→O | | Avg. | |
|---|---|---|---|---|---|---|---|---|---|---|
| | HTER(%) | AUC(%) | HTER(%) | AUC(%) | HTER(%) | AUC(%) | HTER(%) | AUC(%) | HTER(%) | AUC(%) |
| FGHV (Liu et al., 2022b) | 9.17 | 96.92 | 12.47 | 93.47 | 16.29 | 90.11 | 13.58 | 93.55 | 12.88 | 93.51 |
| GDA (Zhou et al., 2022b) | 9.20 | 98.00 | 12.20 | 93.00 | 10.00 | 96.00 | 14.40 | 92.60 | 11.45 | 94.90 |
| PatchNet (Wang et al., 2022a) | 7.10 | 98.46 | 11.33 | 94.58 | 13.40 | 95.67 | 11.82 | 95.07 | 10.91 | 95.95 |
| SSAN (Wang et al., 2022c) | 6.67 | 98.75 | 10.00 | 96.67 | 8.88 | 96.79 | 13.72 | 93.63 | 9.82 | 96.46 |
| IADG (Zhou et al., 2023) | 5.41 | 98.19 | 8.70 | 96.40 | 10.62 | 94.50 | 8.86 | 97.14 | 8.40 | 96.56 |
| UDG-FAS (Liu et al., 2023) | 5.95 | 98.47 | 9.82 | 96.76 | 5.86 | 98.62 | 10.97 | 95.36 | 8.15 | 97.30 |
| TTDG (Zhou et al., 2024) | 4.16 | 98.48 | 7.59 | 98.18 | 9.62 | 98.18 | 10.00 | 96.15 | 7.84 | 97.75 |
| SA-FAS (Sun et al., 2023) | 5.95 | 96.55 | 8.78 | 95.37 | 6.58 | 97.54 | 10.00 | 96.23 | 7.83 | 96.42 |
| DIVT-M (Liao et al., 2023) | 2.86 | 99.14 | 8.67 | 96.92 | 3.71 | 99.29 | 13.06 | 94.04 | 7.08 | 97.35 |
| GAC-FAS (Le & Woo, 2024) | 5.00 | 97.56 | 8.20 | 95.16 | 4.29 | 98.87 | 8.60 | 97.16 | 6.52 | 97.19 |
| FLIP (Srivatsan et al., 2023) | 4.95 | 98.11 | 0.54 | 99.98 | 4.25 | 99.07 | 2.31 | 99.63 | 3.01 | 99.20 |
| CFPL (Liu et al., 2024) | 1.43 | 99.28 | 2.56 | 99.10 | 5.43 | 98.41 | 2.50 | 99.42 | 2.98 | 99.05 |
| I-FAS (Zhang et al., 2025) | **0.32** | 99.88 | 0.04 | 99.99 | 3.22 | 98.48 | **1.74** | **99.66** | 1.33 | 99.50 |
| **Ours** | 0.42 | **99.92** | **0.00** | **100.00** | **1.00** | **99.83** | 2.81 | 99.63 | **1.06** | **99.85** |

## C.2 Fine-grained Analysis on Spoof Type Robustness

To examine whether FaceCoT introduces bias toward certain spoof types, we conduct a fine-grained analysis on the Rose-Youtu (Li et al., 2018), which contains seven representative spoofing attack types. We report the per-type detection accuracy before and after fine-tuning with FaceCoT. As shown in Table 8, our approach achieves consistent improvements across all attack categories. These results indicate that FaceCoT provides more comprehensive semantic supervision and enhances general spoof detection capability, rather than overfitting to the dominant categories in the training set.

Table 8: Performance comparison across different spoof types in the Rose-Youtu test set before and after SFT with FaceCoT.

| Cheat Type | Meaning | Number | Acc. (Zero-shot) | Acc. (After SFT) | Change |
|---|---|---|---|---|---|
| Mc | Mask: Cut eyes & mouth | 202 | 100.00% | 100.00% | – |
| Mf | Mask: Full face | 100 | 74.00% | 100.00% | +26.00% |
| Mu | Mask: Upper part cut | 198 | 93.43% | 100.00% | +6.57% |
| Pq | Printed paper (quivering) | 200 | 0.00% | 95.50% | +95.50% |
| Ps | Printed paper (still) | 200 | 0.00% | 68.00% | +68.00% |
| Vl | Video (Lenovo LCD) | 201 | 0.00% | 96.02% | +96.02% |
| Vm | Video (Mac LCD) | 199 | 0.00% | 71.36% | +71.36% |

## C.3 Comparison of Zero-shot and CoT-trained Models

To assess the effectiveness of our supervised CoT training, we first establish a zero-shot baseline, where large vision-language models are directly prompted with natural language to classify real versus spoof images without any fine-tuning. To further validate the robustness of our approach, we also conduct the same experiment with Qwen2.5-VL (Bai et al., 2025), a recent advanced multimodal VLM. As shown in Table 9, both VLMs perform worse than the SOTA method I-FAS (Zhang et al., 2025) under the zero-shot setting. After applying our CoT-based fine-tuning, we observe consistent and substantial gains on both models: MiniCPMV achieves a reduction of 11.61% in HTER and an improvement of 10.45% in AUC, while Qwen2.5-VL shows similar improvements (HTER reduced by 10.28% and AUC improved by 11.97%). These results demonstrate that our FaceCoT dataset and CEPL framework provide stable and significant benefits across different VLM architectures, enabling stronger discriminative ability and cross-domain generalization than relying on zero-shot reasoning alone.

Table 9: Comparison between zero-shot baselines and our supervised CoT method across different backbone VLM models.

| Method | Average HTER (%) | Average AUC (%) |
|---|---|---|
| I-FAS (Zhang et al., 2025) | 11.31 | 93.71 |
| Zero-shot(Qwen2.5-VL-7B (Bai et al., 2025)) | 19.60 | 83.75 |
| Zero-shot(MiniCPMV-2.6-8B (Yao et al., 2024)) | 17.91 | 87.32 |
| Ours(Qwen2.5-VL-7B (Bai et al., 2025)) | 9.32 ($\downarrow$10.28) | 95.72 ($\uparrow$11.97) |
| Ours(MiniCPMV-2.6-8B) | 6.30 ($\downarrow$11.61) | 97.77 ($\uparrow$10.45) |

## C.4 The effect of Reinforcement Learning on CoT annotation quality

By introducing RL into the training of the caption model, the annotation accuracy is effectively improved. Although this metric demonstrates the effectiveness of RL in enhancing conclusion label accuracy, relying solely on conclusion accuracy is not sufficient to fully evaluate the semantic quality of the generated annotations. To further verify the advantages of RL-generated CoT annotations in terms of linguistic coherence and logical consistency, we design an experiment in which we treat two sets of generated CoT annotations—those produced by a model trained solely via Supervised Fine-Tuning (SFT) and those produced by an SFT-trained model further refined with RL—as separate training sets under our proposed CoT-Enhanced Progressive Learning (CEPL) framework. The comparative results, reported in Table 10, demonstrate the tangible benefits of RL on CoT data quality: RL not only enhances annotation accuracy, but also significantly improves the consistency, coherence, and semantic reliability of the generated CoT explanations.

Table 10: The effect of Reinforcement Learning (RL) on CoT annotation quality: Supervised Fine-Tuning (SFT) versus SFT with RL

| Training methods | Results | |
|---|---|---|
| | HTER(%) | AUC(%) |
| SFT | 8.00 | 96.97 |
| SFT + RL | **6.87** | **97.27** |

Table 11: Ablation study on CoT data at a resolution of $224 \times 224$, with comparison to a setup using only binary label data.

| Data type | Results | |
|---|---|---|
| | HTER(%) | AUC(%) |
| Label | 17.07 | 90.42 |
| Label + CoT | **11.28** | **94.05** |

### C.5 THE EFFECT OF FACECOT UNDER LOW-RESOLUTION

In the ablation study presented in the main text, we evaluate the impact of FaceCoT data using an input resolution of $448 \times 448$. Given that most existing FAS methods (Zhang et al., 2025; Srivatsan et al., 2023; Huang et al., 2022) conduct experiments at a resolution of $224 \times 224$, we perform an ablation study at this resolution to verify the effectiveness of our CoT data under low-resolution settings. We compare a single-stage training regime using only label classification data against a single-stage joint training regime incorporating CoT data. As shown in Table 11, the model trained with CoT data achieves a 6.70% reduction in HTER and a 4.61% increase in AUC. These findings can be summarized as follows: (1) The relative gain at $224 \times 224$ (–5.79% HTER, +3.63% AUC) is substantially larger than at $448 \times 448$ (–1.42% HTER, +1.54% AUC), demonstrating that our CoT annotations help the model recover fine-grained facial cues that are otherwise lost at lower resolutions. (2) Even when applied in a simple single-stage joint training regime, the CoT-augmented model already outperforms current state-of-the-art methods, demonstrating its superior generalization and robustness conferred by CoT data training.

## D USAGE OF LLMs

In this work, Large Language Models (LLMs) were employed as auxiliary tools in two aspects:

- **Manuscript Refinement:** LLMs were used to assist in language polishing and grammar checking after the human authors had completed the technical writing. The scientific content, experiment design, and analysis were fully conducted by the authors.

- **Annotation of FaceCoT-Gold100K:** GPT-4o was used to generate Chain-of-Thought (CoT) annotations. Specifically, we carefully designed prompts to guide the model toward describing spoof-related visual cues (e.g., reflection artifacts, cutting marks), rather than allowing free-form generation. The model outputs were subsequently reviewed and refined by human experts to ensure accuracy, fairness, and domain relevance.

## E LIMITATION AND FEATURE WORK

We have included as many spoofing types as practically possible in FaceCoT to ensure broad coverage. Some uncommon real-world variations in devices or environments are inevitably not captured, which we leave for future extension. Furthermore, while our work primarily emphasizes the utility of FaceCoT for model training and performance gains on downstream FAS tasks, we acknowledge that its potential as a standardized benchmark for evaluation has not been fully explored. In particular, since FaceCoT provides reasoning chains, it can serve as a valuable resource to assess not only predictive accuracy but also the interpretability and coherence of model outputs. We leave a more systematic investigation of FaceCoT's role in model evaluation and benchmarking as an important direction for future work.

## F DEMONSTRATION OF RESULT INTERPRETABILITY

### F.1 DEMONSTRATION OF REAL FACE

Since FAS is inherently a binary classification problem, explaining why a face is real is as important as explaining why a face is spoof. To this end, we present a representative real face case (Fig 11) to illustrate how the model perceives authenticity. Rather than relying solely on the absence of

spoofing cues, the model proactively identifies positive evidence, including natural and proportionate facial structure, realistic skin texture consistent with illumination, and coherence between facial appearance and the surrounding environment. This unified reasoning pipeline ensures that both real and spoof faces are explained under the same framework, with real cases supported by explicit authenticity cues.

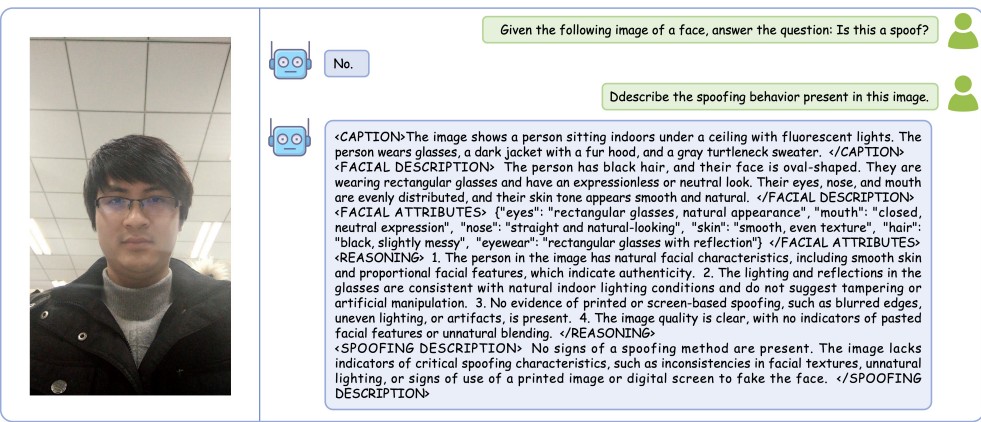

Figure 11: An interpretable result case on a real face

## F.2 DEMONSTRATION OF ELEVEN BENCHMARK DATASET

In this section, we present examples of our model's interpretable outputs on eleven benchmark datasets, including MSU-MFSD Wen et al. (2015), CASIA-MFSD Zhang et al. (2012), Idiap Replay Attack Chingovska et al. (2012), OULU-NPU Boulkenafet et al. (2017), SIW Liu et al. (2018b), Rose-Youtu Li et al. (2018), HKBU-MARs-V1+ Liu et al. (2018a), WMCA George et al. (2019), SIW-M-V2 Guo et al. (2022), CASIA-SURF-3DMask Yu et al. (2020), and HiFiMask Liu et al. (2022a).

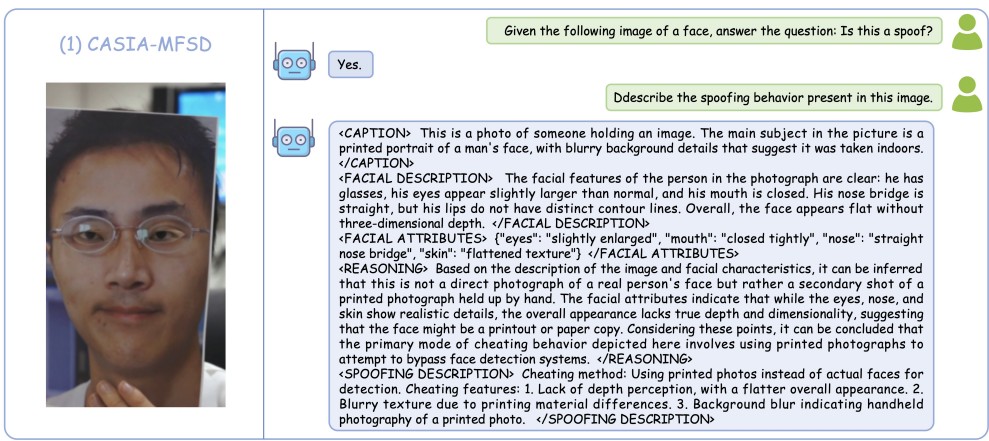

(a) An interpretable result case on CASIA-MFSD

Figure 12: Interpretable CoT outputs on eleven benchmarks

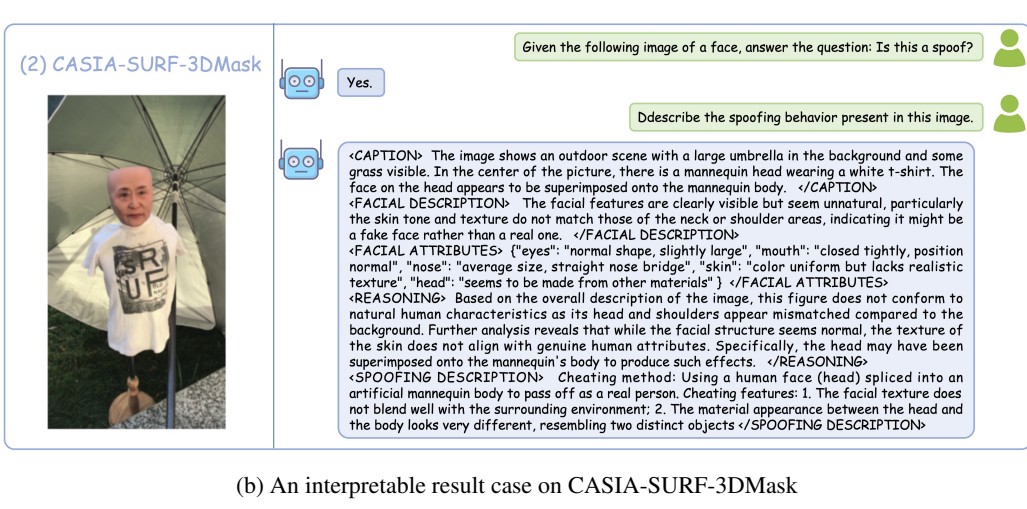

(b) An interpretable result case on CASIA-SURF-3DMask

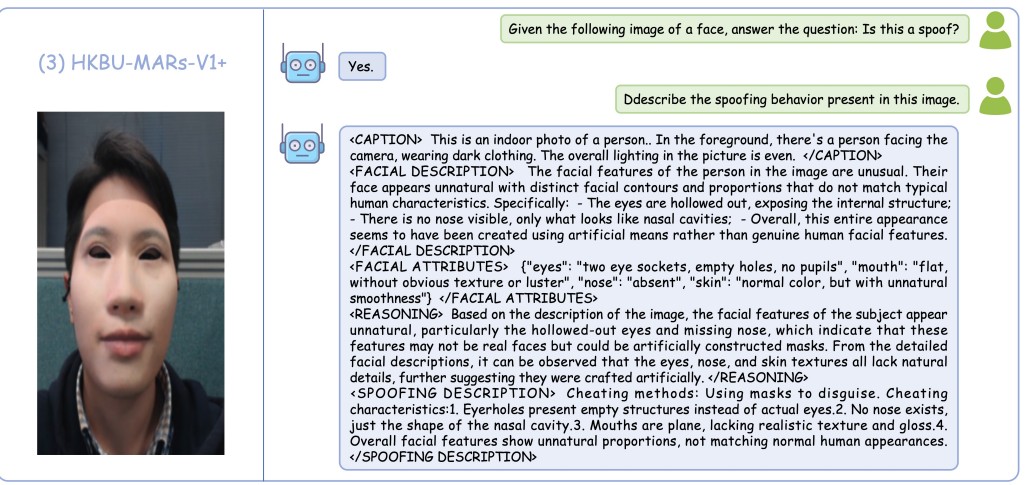

(c) An interpretable result case on HKBU-MARs-V1+

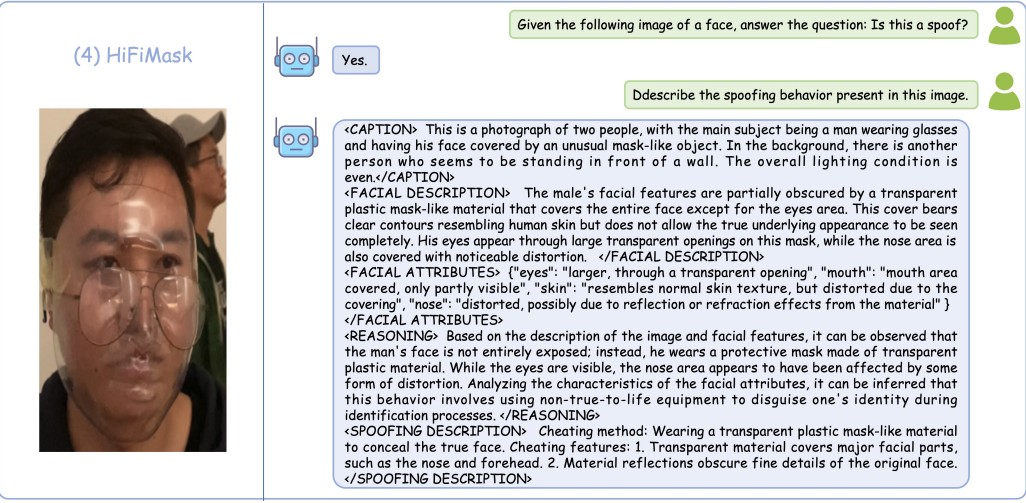

(d) An interpretable result case on HiFiMask

Figure 12: Interpretable CoT outputs on eleven benchmarks

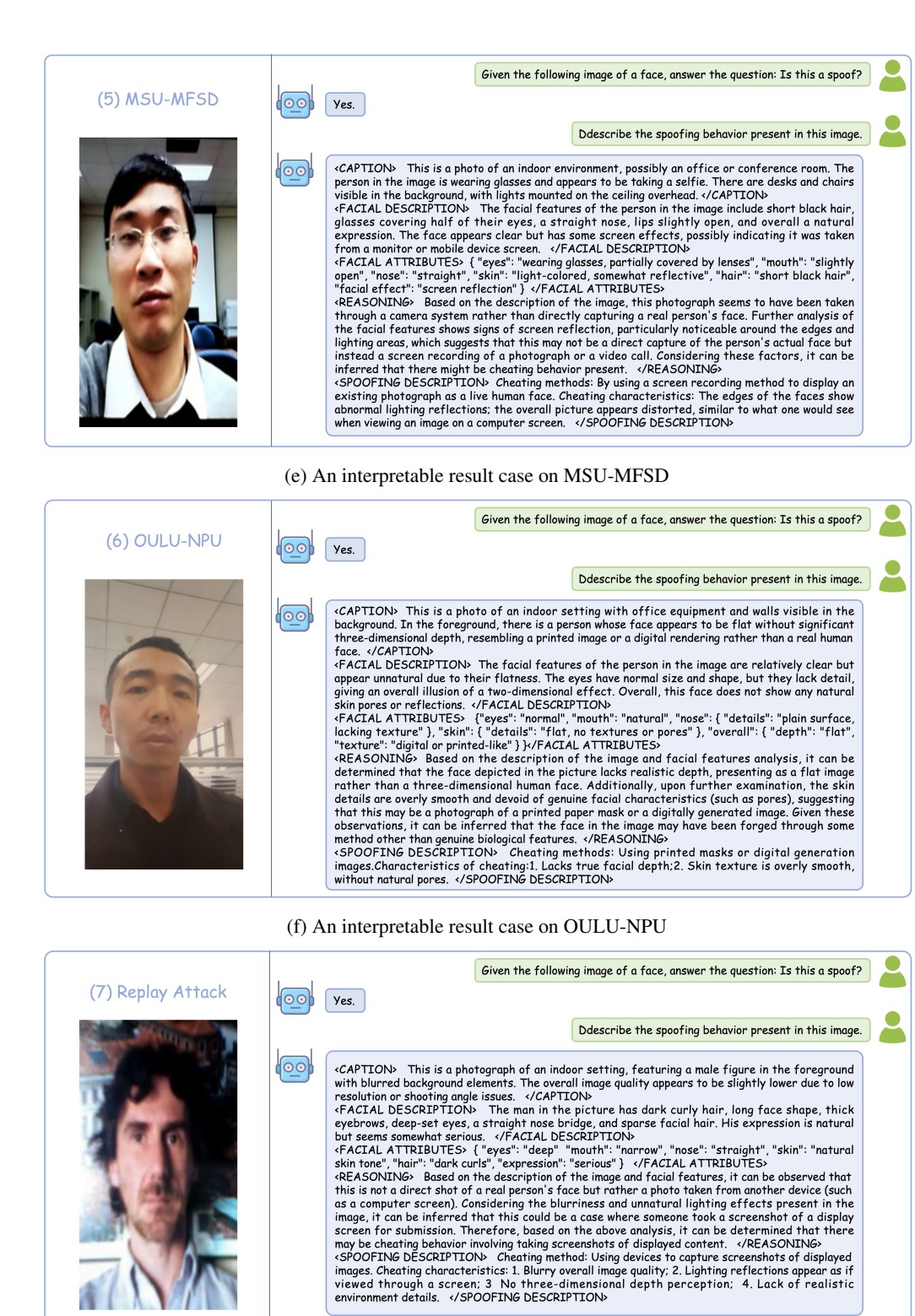

(e) An interpretable result case on MSU-MFSD

(f) An interpretable result case on OULU-NPU

(g) An interpretable result case on Replay Attack

Figure 12: Interpretable CoT outputs on eleven benchmarks

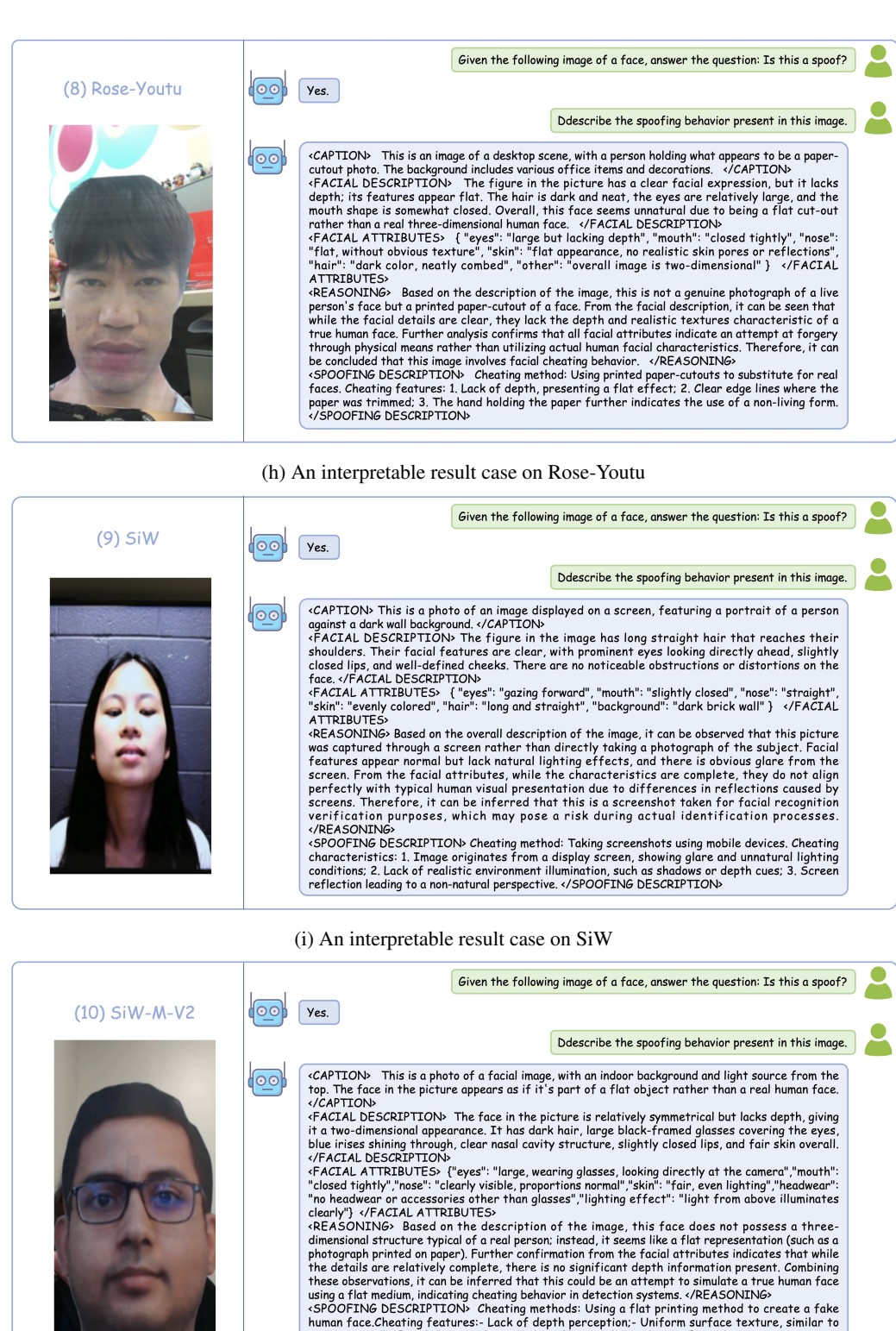

(h) An interpretable result case on Rose-Youtu

(i) An interpretable result case on SiW

(j) An interpretable result case on SiW-M-V2

Figure 12: Interpretable CoT outputs on eleven benchmarks

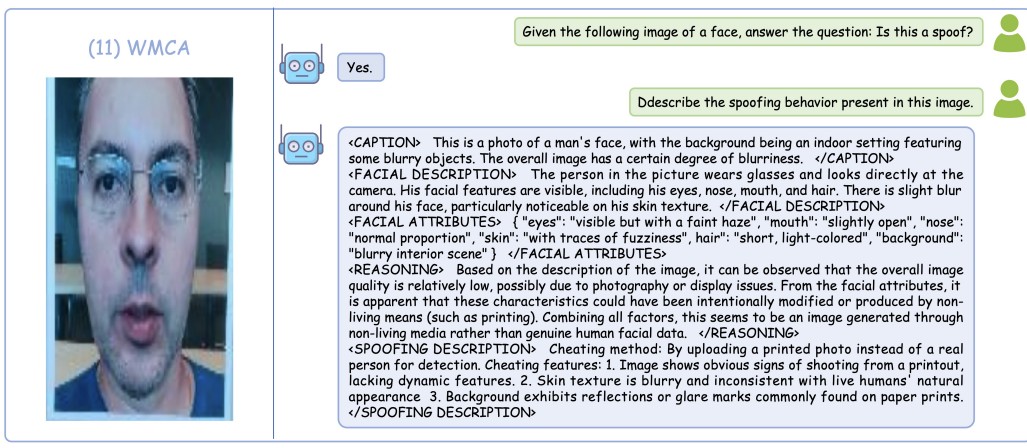

(k) An interpretable result case on WMCA

Figure 12: Interpretable CoT outputs on eleven benchmarks

