# OpenReview forum: "FaceCoT: A Comprehensive Benchmark for Face Anti-Spoofing with Chain-of-Thought Reasoning"
_ICLR.cc/2026/Conference — ICLR 2026 Conference Withdrawn Submission_

### Official Review · Reviewer_uuxn · 2025-10-24

**Soundness:** 3
**Presentation:** 3
**Contribution:** 2
**Rating:** 2
**Confidence:** 4

**Summary:**

This paper introduces FaceCoT, a multimodal VQA dataset and CoT reasoning framework for face anti-spoofing task. The dataset contains 1.08 million samples across 14 attack types, including a verified subset (FaceCoT-Gold100K) and a reinforcement learning–expanded set (FaceCoT-Silver982K). The authors further propose CEPL, a two-stage training strategy: (1) visual enhancement pre-training using CoT data, and (2) multi-task joint training that combines CoT reasoning with binary classification. Experiments on 11 public FAS benchmarks show clear performance gains, demonstrating the effectiveness and generalization ability of the proposed method.

**Strengths:**

The paper addresses a clear and forward-looking problem. Introducing CoT reasoning into face anti-spoofing is novel and enhances model interpretability.
The FaceCoT dataset is large and diverse, offering rich vision-language annotations and a strong basis for future research.
The proposed CEPL framework adopts a clear two-stage training strategy that effectively balances reasoning and classification, improving both accuracy and interpretability.

**Weaknesses:**

The paper approaches the problem from the perspective of Chain-of-Thought (CoT) and interpretability, yet all evaluations of interpretability are presented as qualitative descriptions, lacking quantitative metrics or user studies to assess the usefulness or accuracy of the explanations.

The multi-task joint training lacks rigorous empirical validation. The authors emphasize the importance of MJT, but no ablation comparing “VEP-only” or removing MJT is provided, making it unclear whether the multi-task mechanism itself contributes to the improvement.

The choice of LoRA fine-tuning lacks experimental justification. In CEPL, the first stage uses full-parameter fine-tuning, while the second stage adopts LoRA. Although the authors claim it prevents catastrophic forgetting, there is no quantitative comparison or verification, making this conclusion largely speculative.

The consistency between CoT reasoning and final predictions is not validated. The evaluation relies solely on binary classification metrics, which cannot demonstrate whether the generated reasoning aligns with the final decision or whether CoT supervision truly contributes to model reasoning behavior.

**Questions:**

See weakness

---

### Official Review · Reviewer_HErr · 2025-10-26

**Soundness:** 2
**Presentation:** 3
**Contribution:** 2
**Rating:** 4
**Confidence:** 4

**Summary:**

This paper addresses the core issues of poor generalization and interpretability of unimodal methods in the face anti-forgery (FAS) task by proposing the FaceCoT dataset and the accompanying CEPL training strategy. FaceCoT is the first large-scale visual question answering dataset for FAS, containing 1.08 million samples, covering 14 attack types, and innovatively introducing a six-layer structure called Chain of Thought (CoT) annotation. The CEPL strategy effectively improves model performance through visually enhanced pre-training and multi-task joint training. Experiments show that this method achieves state-of-the-art performance on 11 benchmark datasets (with an average AUC improvement of 4.06%). However, the paper lacks significant verification of the method's innovation and experimental depth.

**Strengths:**

1. Data Scale and Quality: 1.08 million samples, covering 14 attack types, with annotation quality ensured through manual verification and reinforcement learning optimization (GPT-4o + manual verification + RL optimization).
2. Accurate Problem Identification: Targets the bottlenecks of generalization and interpretability in the FAS domain, proposing a multimodal solution that points in the right direction.
3. Extensive Experimental Coverage: Tested on 11 benchmark datasets, demonstrating strong cross-domain performance.
4. Extensive Explanation of Interpretability: The appendix provides extensive CoT output examples to enhance the credibility of the results.

**Weaknesses:**

1. Lack of innovation in dataset construction methods:
The RL annotation model directly uses the existing VRFT framework without any improvements tailored to the specific characteristics of the FAS task.
Compared to works such as ForgerySleuth (Sun et al., 2024) and BusterX (Wen et al., 2025), the methodological advantages are not demonstrated.
2. Validation of visual encoder improvements is lacking:
Feature visualization comparison before and after pre-training is lacking.
No feature space analysis is performed to demonstrate the effectiveness of fine-grained feature learning, and no corresponding ablation experiments are performed.
3. Gaps in CoT module contribution analysis:
The necessity and contribution of each module in the six-layer CoT architecture are not verified.
Unable to distinguish whether the performance improvement is due to the CoT design or a simple data scale effect.
4. Incomplete baseline comparison:
Lack of comparison with FAS methods based on open-source MLLMs such as LLaVA.
No significant advantages over simple baselines are demonstrated.

**Questions:**

1. Innovation question: Compared with recent work such as ForgerySleuth and BusterX, what substantial innovations does this paper offer in terms of RL annotation model design and multimodal fusion methods? Please provide a detailed comparative analysis.
2. Feature Learning Verification: How can we prove that the visual encoder indeed learns better fine-grained features through CoT supervision? Please provide direct evidence such as feature visualization and attention analysis.
3. CoT Architecture Necessity: Are all six layers of CoT modules indispensable? Please provide module-level ablation experiments to quantify the contribution of each module to the final performance.
4. Method Advantage Verification: Compared with directly applying existing MLLMs (such as LLaVA) to the FAS task, what are the advantages of this method? Please provide corresponding comparative experiments.

---

### Official Review · Reviewer_eFWt · 2025-10-31

**Soundness:** 2
**Presentation:** 2
**Contribution:** 2
**Rating:** 4
**Confidence:** 5

**Summary:**

The paper presents FaceCoT, a large CoT-annotated resource for face anti-spoofing (FAS), and a two-stage training scheme called CoT-Enhanced Progressive Learning (CEPL). Stage-1 performs full-parameter SFT on CoT text (termed “Visual Enhancement Pre-training”), and Stage-2 conducts multi-task joint training for CoT generation and binary classification while inheriting the Stage-1 vision encoder. The approach reports strong cross-domain results on 11 benchmarks and showcases interpretable CoT outputs.

**Strengths:**

1.FaceCoT dataset: A large, structured CoT dataset tailored to FAS (three major attack types, 14 subtypes) with a clear six-part template (Caption, Facial Description, Facial Attributes, Reasoning, Spoofing Description, Conclusion).
2.Methodological advantage: CEPL’s stage-wise optimization reduces task interference compared to single-stage multi-task training.
3.Breadth of empirical validation: The method is evaluated across 11 public FAS datasets, achieving the best scores and highlighting robustness in cross-domain settings.

**Weaknesses:**

1.Stage-1 motivation vs. evidence attribution.
Stage-1 is described as “Visual Enhancement” aimed at strengthening the vision encoder, yet the implementation performs full-parameter SFT (vision encoder, connector/resampler, and LLM all updated). Without module-freezing or representation probes, improvements cannot be causally ascribed to the vision side (language-side learning or template alignment may also drive gains). The text and Fig. 4 explicitly present this setup, but current ablations (CEPL vs. single-stage; with/without RL) do not isolate the source of the benefit. More controlled analyses are needed to substantiate the “visual representation” claim.
2.Potential shortcut / label-template leakage from category-specific hints.
The paper provides tailored hints per spoofing type (e.g., “Photographing a printed photo constitutes spoofing.”), and also enforces exact match of <Conclusion> to the ground truth during verification. This design risks a shortcut from hint→label rather than genuine visual analysis. Clarification is needed on whether such hints are visible to the detector during training (beyond the caption-model annotation phase) and how performance behaves without hints.
3.Cross-domain evaluation lacks finer-grained decomposition.
The paper emphasizes robustness to unseen attack modalities (e.g., HKBU-MARs-V1+ and HiFiMask include transparent, plaster, and resin masks) and shows strong overall gains. However, it does not provide per-attack-type breakdowns and significance tests. Per-attack-type granular evaluations are needed to verify whether improvements are uniform across subtypes.
4.Instructional baselines and CoT ablations are not yet sufficient.
To separate the value of task instruction from the value of the multi-field CoT structure, include an instructional baseline that asks only the task question (no intermediate CoT fields). This will clarify whether CoT structure—not just task prompting—drives the observed gains.

**Questions:**

1 Stage-1 claims to enhance visual representations, but it updates all modules. Can you provide controlled evidence (e.g., module freezing) that attributes gains to the vision encoder rather than the language side?
2 Precisely when/where are hints injected? Are they present during detector training or only in the caption-model pipeline? Please report with-hint vs. no-hint performance.
3 Please add an instruction-only baseline (only the task question; no intermediate fields) to quantify each field’s contribution.
4 For cross-domain claims on unseen attacks (e.g., transparent/plaster/resin masks in HKBU-MARs-V1+ and HiFiMask), provide per-attack-type results with dispersion and significance to substantiate uniform gains.

---

### Official Review · Reviewer_nPFN · 2025-10-31

**Soundness:** 3
**Presentation:** 2
**Contribution:** 2
**Rating:** 4
**Confidence:** 4

**Summary:**

This paper introduces a dataset called FaceCoT, the first visual question-answering dataset specifically designed for face anti-spoofing (FAS), containing 14 attack types as well as structured CoT annotations. Besides, the paper proposes a CoT-Enhanced Progressive Learning strategy (CEPL), which optimizes the visual encoder and performs multi-task joint optimization through staged training. Experimental results show the good performance of the proposed method.

**Strengths:**

1.This paper is easy to follow.

2.This paper presents an open-sourced well large scale of VQA dataset for FAS task, which will benefit the FAS research community.

3.The proposed CoT structure with CEPL strategy sounds reasonable and improves the reasoning and detection performance.

**Weaknesses:**

1.The technical contribution of this paper is limited, as the basic SFT and RL are conventional without specific design for FAS task.

2.During the data expansion, unlabeled spoofing data is utilized to perform RL of the caption model. With simple format and accuracy rewards, it is hard to handle semantic errors, as well as the unseen spoofing types, which raises my concerns about the accuracy of the expended annotation.

**Questions:**

See Weakness.

---

### Note · Authors · 2025-11-12

I have read and agree with the venue's withdrawal policy on behalf of myself and my co-authors.